# In Vitro Skin Permeation of Idebenone from Lipid Nanoparticles Containing Chemical Penetration Enhancers

**DOI:** 10.3390/pharmaceutics13071027

**Published:** 2021-07-06

**Authors:** Lucia Montenegro, Ludovica Maria Santagati, Maria Grazia Sarpietro, Francesco Castelli, Annamaria Panico, Edy Angela Siciliano, Francesco Lai, Donatella Valenti, Chiara Sinico

**Affiliations:** 1Department of Drug and Health Sciences, University of Catania, 95125 Catania, Italy; ludovica.santagati@icloud.com (L.M.S.); mg.sarpietro@unict.it (M.G.S.); fcastelli@unict.it (F.C.); panico@unict.it (A.P.); edysiciliano@hotmail.it (E.A.S.); 2Department of Life and Environmental Sciences, University of Cagliari, 09126 Cagliari, Italy; frlai@unica.it (F.L.); valenti@unica.it (D.V.); sinico@unica.it (C.S.)

**Keywords:** idebenone, chemical penetration enhancers, lipid nanoparticles, skin permeation, SLN, NLC

## Abstract

Lipid nanoparticles (LNPs) have been proposed as carriers for drug skin delivery and targeting. As LNPs effectiveness could be increased by the addition of chemical penetration enhancers (PE), in this work, the feasibility of incorporating PE into LNPs to improve idebenone (IDE) targeting to the skin was investigated. LNPs loading IDE 0.7% *w*/*w* were prepared using hydrophilic (propylene glycol, PG, 10% *w*/*w* or *N*-methylpyrrolidone, NMP, 10% *w*/*w*) and/or lipophilic PE (oleic acid, OA, 1% *w*/*w*; isopropyl myristate, IPM, 3.5% *w*/*w*; a mixture of 0.5% *w*/*w* OA and 2.5% *w*/*w* IPM). All LNPs showed small sizes (<60 nm), low polydispersity index and good stability. According to the obtained results, IDE release from LNPs was not the rate-limiting step in IDE skin penetration. No IDE permeation was observed through excised pigskin from all LNPs, while the greatest increase of IDE penetration into the different skin layers was obtained using the mixture OA/IPM. The antioxidant activity of IDE-loaded LNPs, determined by the oxygen radical absorbance capacity assay, was greater than that of free IDE. These results suggest that the use of suitable PE as LNPs components could be regarded as a promising strategy to improve drug targeting to the skin.

## 1. Introduction

Lipid nanoparticles (LNPs) have drawn a great deal of attention as drug delivery systems for topical administration owing to their advantages over other colloidal carriers, including the occlusive effect, improved skin hydration and increased skin permeation of active ingredients, which may result in greater effectiveness of the entrapped molecules [1,2,3,4,5]. As reported in the literature [6], LNPs occlusive properties may be affected by the degree of crystallinity as well as by the number and size of particles in the vehicle. In particular, several works [7,8,9] highlighted that solid lipid nanoparticles (SLNs) showed greater occlusive properties compared to nanostructured lipid carriers (NLCs) owing to their higher degree of crystallinity. SLNs consist of a solid lipid core stabilized by surfactants in an aqueous medium, while an amorphous structure and/or the presence of liquid lipids are typical of the lipid core of NLCs. The highly ordered arrangement of the lipid matrix account for SLNs higher degree of crystallinity, although it could lead to drawbacks such as poorer stability due to drug leakage from the nanocarrier during preparation and storage and lower drug loading capacity. Therefore, in the last decades, several solid and liquid lipids have been investigated to obtain LNPs with improved technological properties and better therapeutic outcomes after topical application [10].

A strategy that could result in improved drug skin targeting and delivery is the inclusion of chemical penetration enhancers in nanoparticle colloidal suspensions. Such a strategy proved successful in increasing drug transport through the skin from liposomes [11], but, to date, few data have been reported about the ability of chemical penetration enhancers to improve drug skin penetration/permeation from LNPs. Transcutol P, a hydrophilic chemical penetration enhancer, has been reported to increase skin accumulation of 8-methoxypsoralen from LNPs [12]. Patel et al. [13] added lipophilic penetration enhancers (d-limonene and oleic acid) to gel formulations containing raloxifene-loaded solid lipid nanoparticles, showing that the addition of d-limonene provided an increase of drug permeation through excised human skin.

Therefore, in this work, we investigated the feasibility of using lipophilic and/or hydrophilic penetration enhancers as components of LNPs colloidal suspensions to improve drug skin targeting and delivery.

Idebenone (IDE), a synthetic antioxidant analogous of coenzyme Q_10_, was chosen as the model drug on the grounds that in a previous paper, SLNs loaded with IDE provided an increase of drug penetration in the upper skin layers compared to free IDE, thus highlighting a targeting effect due to drug entrapment into such nanocarriers [14].

Two lipophilic penetration enhancers, namely oleic acid (OA) and isopropyl myristate, were selected as oily ingredients to be incorporated into the lipid core of NLCs, owing to their safety and well-documented effectiveness in promoting skin penetration/permeation of a great variety of active compounds [15,16,17,18].

Regarding the hydrophilic penetration enhancers, the choice fell on propylene glycol (PG) and *N*-methylpyrrolidone (NMP), two of the most widely investigated penetration enhancers, which are currently used both in pharmaceutical and cosmetic fields [15,16,17].

## 2. Materials and Methods

### 2.1. Materials

Glyceryl oleate (Tegin O^®^, GO), isopropyl myristate (IPM), cetyl palmitate (CP) and imidazolidinyl urea (Kemipur 100^®^) were bought from ACEF (Fiorenzuola D’Arda, Italy). Idebenone (IDE, purity: minimum 98% as reported in the compound data sheet) was obtained from Carbosynth (Berkshire, UK). Polyoxyethylene-20-oleyl ether (Brij 98^®^, Oleth-20) was obtained from Farmalabor (Canosa di Puglia, Italy). Oleic acid (OA), *N*-methylpyrrolidone (NMP) and propylene glycol (PG) were obtained from Sigma Aldrich (Milan, Italy). Poloxamer 188 (Lutrol^®^F68) was a gift from BASF (Ludwigshafen, Germany). Regenerated cellulose membranes (Spectra/Por CE; Mol. Wt. Cut off 3000) were bought from Spectrum (Los Angeles, CA, USA). Solvents (methanol and water) used in the HPLC procedures were of LC grade and were obtained from Merck (Darmstadt, Germany). All other reagents were of analytical grade and used as received.

### 2.2. Preparation of Lipid Nanoparticles

Lipid nanoparticles’ composition is illustrated in Table 1. All carriers were prepared by the phase inversion temperature (PIT) method, as previously reported [19,20]. Unloaded and drug-loaded lipid nanocarriers were prepared using the same procedure. The aqueous phase (deionized water) contained Kemipur 100^®^ 0.35% *w*/*w* as preservatives with and without the hydrophilic penetration modifiers NMP (10.0% *w*/*w*) or PG (10.0% *w*/*w*). NMP and PG were used at a concentration (10% *w*/*w*) that is generally regarded as effective for both enhancers [15]. IDE (0.7% *w*/*w*) was added to the oil phase consisting of oleth-20, glyceryl oleate and different percentages of cetyl palmitate (CP) with and without lipophilic penetration modifiers (oleic acid and isopropyl myristate). Preliminary experiments were performed to determine the greatest concentration of lipophilic enhancers that provided stable LNPs. According to the results of such experiments, IPM was used at 3.5% *w*/*w* while OA was incorporated at 1% *w*/*w*. The amount of CP used to prepare LNPs changed to maintain the total amount of lipids in the LNPs matrix constant. After heating at 90 °C the oil and aqueous phase separately, the aqueous phase was added dropwise to the oil phase under stirring (700 rpm). While cooling down to 25 °C, the colloidal suspension turned clear at the phase inversion temperature (PIT), and the PIT value was recorded using a conductivity meter (model 525, Crison, Modena, Italy). Lipid nanoparticle samples were stored in airtight jars at room temperature and sheltered from light until used.

### 2.3. Transmission Electron Microscopy (TEM)

TEM was performed using a transmission electron microscope (model JEM 2010, Jeol, Peabody, MA, USA) operating at an acceleration voltage of 200 KV. Colloidal suspensions of lipid nanoparticles (5 μL) were placed on a Formvar (200-mesh) copper grid (TAAB Laboratories Equipment, Berks, UK). After sample absorption, the surplus was removed by filter paper, and an aqueous solution of uranyl acetate (2% *w*/*w*) was added. The sample was allowed to dry at 25 °C, and TEM images were acquired.

### 2.4. Dynamic Light Scattering (DLS)

Particle size and polydispersity index (PDI) of lipid nanoparticles were determined by DLS using a Zetasizer Nano ZS90 (Malvern Instruments, Malvern, UK) with a 4-mW laser diode at 670 nm and scattering light at 90°. All samples were analyzed after dilution (1:5, sample/distilled water) at 25 °C. Each measurement was carried out in triplicate and the results were expressed as mean ± SD.

ζ-potential was obtained by laser Doppler velocimetry using the same Zetasizer. Sample dilution was performed using KCl 1 mM (pH 7.0) prior to the analysis [21].

### 2.5. Differential Scanning Calorimetry (DSC)

DSC analyses of lipid nanoparticles were carried out using a Mettler TA STAR^e^ instrument (Mettler Toledo, Greifensee, Switzerland) equipped with a DSC 822^e^ cell and Mettler STAR^e^ V8.10 software. As reference, 100 μL of deionized water containing 0.35% *w*/*w* imidazolidinyl urea was used.

The instrument was calibrated using indium and palmitic acid (purity ≥99.95% and ≥99.5%, respectively; Fluka, Buchs, Switzerland). A 160 μL calorimetric pan was filled with 100 µL of each lipid nanoparticle sample, and DSC analyses were performed, heating the sample from 5 to 65 °C (rate 2 °C/min) and then cooling from 65 to 5 °C (rate 4 °C/min), for at least three times to test the reproducibility of thermodynamic parameters. All measurements were carried out in triplicate.

### 2.6. Stability Tests

Stability studies were performed, determining particle sizes, PDI and ζ-potential values of each colloidal suspension at regular intervals (24 h, one week, one month, three months) during storage at room temperature and in the dark.

### 2.7. In Vitro Release Experiments

IDE release rate from the LPNs under investigation was determined through regenerated cellulose membranes using Franz-type diffusion cells (LGA, Berkeley, CA, USA). The reliability of this method to perform in vitro drug release studies on topical formulations has been reported since 1989 [22]. Prior to the experiments, the cellulose membranes were immersed in deionized water for 1 h at room temperature. Then, they were placed in Franz-type diffusion cells (diffusion surface area 0.75 cm^2^, receptor volume 4.5 mL). A mixture of water/ethanol (50/50, *v*/*v*) was used as receiving phase to ensure pseudo-sink conditions by increasing IDE solubility. This type of receiving phase has already been used for IDE in vitro release studies from SLN, resulting in no significant change of nanoparticle integrity [14]. The receiving solution was constantly stirred (700 rpm) and thermostated at 35 °C to maintain the membrane surface at 32 °C in order to mimic the temperature of the skin surface. Samples of each formulation (200 μL) were applied on the membrane surface for 24 h. All experiments were carried out under non-occlusion conditions and sheltered from light. Samples (200 μL) of the receiving phase were withdrawn at intervals (0, 1, 2, 4, 6, 8 and 24 h), and replaced with the same volume of receptor medium pre-thermostated to 35 °C. The collected samples were analyzed by HPLC to determine the amount of IDE released. After 24 h, the LNPs applied on the membrane surface were analyzed to determine their technological properties (mean particle sizes, PDI and ζ-potential). All experiments were carried out in triplicate.

### 2.8. In Vitro Skin Penetration Experiments

In vitro penetration experiments were performed by means of Franz-type diffusion cells with an effective diffusion area of 0.75 cm^2^, using skin fragments excised from newborn pigs (Goland–Pietrain hybrid pigs, ∼1.2–1.5 kg, died by natural causes and provided by a local slaughterhouse). After careful removal of the subcutaneous fat, the skin was cut into squares of 3 cm × 3 cm and stored at −80 °C until used. Skin samples were pre-equilibrated in normal saline at 25 °C, 2 h before the experiments. Skin specimens were sandwiched securely between the donor and receptor compartment of each Franz cell, with the dermis in contact with the receiving phase consisting of 4.5 mL of 5% *w*/*v* Poloxamer 188 water solution. A receptor fluid different from that employed for in vitro release experiments was chosen as the barrier integrity of animal skin could be impaired by the use of a mixture of water/ethanol (50/50, *v*/*v*) [23]. As the design of Franz cells used in in vitro skin permeation experiments was slightly different, the thermostating bath temperature was set at 37 ± 1 °C to obtain the physiological skin temperature (i.e., 32 ± 1 °C). Samples (200 μL) of each formulation under investigation were placed onto the skin surface under non-occlusive conditions. The receiving solution (stirred at 700 rpm) was withdrawn at intervals (0, 1, 2, 4, 6, 8 and 24 h), replaced with an equal volume of pre-thermostated (37 °C) solution and assayed by HPLC to determine the amount of IDE permeated. At the end of the experiment, the skin surface was washed. The stratum corneum (SC) removal was performed by stripping with adhesive tape (Tesa^®^ AG, Hamburg, Germany). The adhesive tape was firmly pressed on the SC and pulled off with a rapid and fluent stroke. The epidermis was separated from the dermis with a surgical scalpel. Tape strips, epidermis and dermis were put individually in methanol. The resulting samples were sonicated to extract IDE, and the methanol extracts were analyzed to determine IDE content by HPLC. The results were expressed as cumulative amount of IDE penetrated into the different skin layers after 24 h. Experiments were carried out in triplicate.

### 2.9. High-Performance Liquid Chromatography (HPLC) Analysis

HPLC analyses were performed using a Hewlett-Packard model 1050 liquid chromatograph (Hewlett-Packard, Milan, Italy) with a 20 μL Rheodyne model 7125 injection valve (Rheodyne, Cotati, CA, USA) and an UV-VIS detector (Hewlett-Packard, Milan, Italy).

A Simmetry, 4.6 cm × 15 cm reverse phase column (C_18_) (Waters, Milan, Italy) was used, and samples were eluted using a mobile phase consisting of methanol/water 80/20 *v*/*v* (flow rate 1 mL/min) at room temperature. IDE detection was performed at 280 nm. A standard calibration curve was constructed by plotting known concentrations of IDE vs. the corresponding peak areas to perform IDE quantification in the samples under investigation (sensitivity 0.1 μg/mL). Formulation components did not interfere with the analytical determination of IDE.

### 2.10. Oxygen Radical Absorbance Capacity (ORAC) Assay

Scavenging activity against the peroxyl radical (ROO•) of free IDE and IDE-loaded LNPs was assessed. The peroxyl radical (ROO•) was generated by thermo-decomposition of 2,2 azobis (2-aminopropane) dihydrochloride 100 mM (AAPH, Sigma-Aldrich srl, Milan, Italy). The assay was based on the measurement of the fluorescence decrease of fluorescein (FL, 10 nM) after its oxidation in presence of AAPH and the investigated samples (properly diluted) by means of a VICTOR Wallac 1420 Multilabel Counters fluorimeter (PerkinElmer, Waltham, MA, USA) (excitation *λ* = 540 nm, emission *λ* = 570 nm) [24,25]. Relative fluorescence units were determined after incubation for six hours at 37 °C, pH 7.0, and each measurement was carried out in triplicate.

### 2.11. Statistical Analysis

Mean values ± standard deviation (S.D.) were calculated, and Student’s *t*-test was used to evaluate the significance of the difference between mean values. Values of *p* < 0.05 were considered statistically significant.

## 3. Results and Discussion

In a previous paper [14], in vitro release and skin permeation of IDE from SLN prepared using different surfactants were investigated. The results of this study pointed out that the greatest amount of IDE into the upper skin layers (stratum corneum and epidermis) was obtained using oleth-20 as the surfactant. Therefore, in this work, to evaluate the ability of hydrophilic and lipophilic penetration enhancers to improve IDE skin targeting, IDE-loaded LNPs were prepared using oleth-20. As illustrated in Table 2, all LNPs showed small particle sizes (17–60 nm), and no relationship between presence of enhancers (hydrophilic or lipophilic) into LNPs colloidal suspensions and size of the resulting nanoparticles could be pointed out. Apart from formulation DA, all LNPs had PDI values lower than 0.300, thus indicating the formation of monodisperse colloidal systems. LNPs ζ-potential values ranged from −7 to −23 mV, but no correlation was observed between the electric surface charge and the type of enhancer incorporated in the colloidal suspension. Although ζ-potential values greater (as absolute value) than 30 mV are regarded as an essential requisite for colloidal suspension stabilization [26], all investigated LNPs proved stable during storage for three months at room temperature and sheltered from light, as no significant change of particles size, PDI and ζ-potential values was detected (data not shown). In previous works on IDE-loaded SLN and NLC [19,20,21], similar low ζ-potential values did not affect the stability of the colloidal suspensions that proved stable up to 12 months. The good stability of the investigated LNPs could be attributed to a steric stabilization due to the presence on the nanoparticle surface of long polyoxyethylene chains of the surfactant (oleth-20) used to prepare such LNPs.

TEM images showed that all LNPs were roughly round-shaped, regardless of the type of enhancer used for their preparation. As similar images were obtained for all LNPs, in Figure 1, only LNPs obtained using IPM and PG or NMP as enhancers (formulations BA, BB, BC) were reported as examples.

The interactions among the components of the lipid core of IDE-loaded LNPs were studied by DSC, and the resulting calorimetric curves are shown in Figure 2. Cetyl palmitate (CP), the solid lipid used to obtain LNPs, showed two main peaks at 39 °C and 50 °C, in accordance with previously reported data [21,27]. IDE endothermic peak was detected at 46 °C and was not present in formulations AA, BA, CA and DA, thus confirming that IDE was incorporated in an amorphous state into the lipid core of these LNPs. SLN AA showed a main peak at 43 °C and a shoulder at higher temperature. SLN AA melting temperature was about 10 °C lower than that of CP, owing to an increase of surface area resulting from LNPs colloidal sizes and to interactions between lipid and surfactant molecules that led to a less ordered stucture [28]. LNPs BA showed a low and broad calorimetric peak with a melting temperature of 35 °C that suggested strong interactions between the liquid lipid IPM and the other LNPs components. On the contrary, LNPs obtained using OA as oily penetration enhancer provided a well-defined peak at 41.5 °C while a shoulder was detected at higher temperature. This behavior could be attributed to the similarity between the chemical structure of OA and the acyl chains of the surfactant oleth-20, which could lead to a more ordered arrangement of the lipid core compared to that in formulation BA. Formulation CA, obtained using a mixture of OA and IPM, exhibited a single peak at about 35 °C, indicating that the simultaneous presence of both enhancers affected the thermal behavior of the LNPs lipid core, resulting in lower interactions between liquid and solid lipid components. The addition of hydrophilic enhancers (PG or NMP) to the aqueous phase of formulations AA, BA, CA and DA did not lead to any significant change of the thermal behavior of the resulting colloidal suspensions (data not shown), thus suggesting that no interactions between hydrophilic enhancer and nanoparticle lipid core occurred.

As drug release from the vehicle could be the rate-liming step in the perceutaneous absorption process, preliminary in vitro release studies were performed to evaluate IDE release from the investigated LNPs. Such studies were carried out using the infinite dose technique to avoid drug depletion from the donor compartment during the experiment that could prevent the achievement of steady-state conditions. As already reported for lipophilic drugs loaded into LNPs [29], IDE release was supposed to occur only from the LNPs lipid core, owing to IDE’s poor water solubility that hindered its solution in the water phase of the colloidal suspension. IDE release profiles, obtained by plotting the amount released in 24 h from the different LNPs as a function of time, are shown in Figure 3.

All LNPs provided similar IDE release patterns independent of the presence of lipophilic or hydrophilic enhancers in the colloidal suspensions. Lower IDE cumulative release after 24 h was observed from formulations AA, BC, CC and DA that contained no enhancer, IPM and PG, mixture OA/IPM with PG and OA alone, respectively. Therefore, no correlation could be hypothesized between the presence and type of enhancer and resulting IDE release from the colloidal suspension. Applying the LNPs under investigation on the skin surface did not provide any IDE permeation, as this drug was not detected in the receiving chamber up to 24 h. A similar behavior has already been observed in a previous work on in vitro IDE skin permeation from SLN [14]. Therefore, the tested chemical penetration enhancers did not affect LNPs ability to target IDE to the upper skin layers. As shown in Table 3, the total amount of IDE penetrated into the different skin layers after 24 h from LNPs was significantly lower than the amount released from the colloidal suspensions after the same period. Consequently, IDE release from LNPs could not account for the lack of IDE permeation through excised skin and for the different IDE skin penetration obtained from LNPs AA–DC. The data reported in Table 3 pointed out that only the mixture of OA/IPM (formulation CA) was able to provide a notable increase of IDE skin penetration compared to formulation AA, while the inclusion of IPM (formulation BA) or OA (formulation DA) into the lipid core resulted in a slight decrease. The incorporation of NMP in the aqueous phase of LNPs provided a decrease of IDE skin penetration from SLN (formulation AB) and from NLC containing the mixture of OA/IPM (formulation CB) while it led to an increase from formulations containing IPM (formulation BB) or OA (formulation DB). The other hydrophilic enhancer, PG, increased IDE skin penetration only when added to the aqueous phase of SLN (formulation AC) while its addition proved unfavorable in all LNPs containing lipophilic enhancers (formulations BC, CC, DC) in comparison to the corresponding LNPs without PG. It is interesting to note that both hydrophilic enhancers, PG and NMP, significantly decreased the enhancement effect of the mixture OA/IPM, suggesting that the simultaneous inclusion of lipophilic and hydrophilic enhancers could be disadvantageous depending on the specific combination of enhancers used.

Regarding IDE skin penetration into the different skin layers (stratum corneum, epidermis and dermis), Figure 4 highlights that all LNPs provided an accumulation of IDE in the upper skin layers (stratum corneum and epidermis) and the greatest increase was obtained when formulation CA was applied on the skin surface.

The addition of NMP to the aqueous phase of the colloidal suspension provided a significant increase (*p* < 0.05) of IDE skin penetration in the dermis when IPM was used as enhancer (formulation BC), while it led to a significant (*p* < 0.05) decrease in all skin layers from formulation AA. On the contrary, the other hydrophilic enhancer, PG, significantly increased (*p* < 0.05) IDE amount in the dermis from all investigated formulations compared to that from formulation AA (formulations AC, BC, CC, DC), thus suggesting a different IDE distribution in the upper skin layers due to the presence of this enhancer in the vehicle. It is noteworthy that IPM and OA were able to increase IDE skin penetration compared to that from formulation AA (SLN with no enhancers) only when they were used in mixture, while their inclusion as single oily component into the lipid nanoparticle core (formulations BA and DA) even reduced IDE skin penetration into the stratum corneum and epidermis. Such findings could be due to unfavorable interactions between IDE and stratum corneum lipids occurring because of IDE entrapment into a differently packed lipid core, as suggested by DSC data reported and discussed above. Additionally, the OA penetration enhancement effect is thought to rely on its ability to partition from the vehicle to the upper skin lipid layer in which it dispersed, promoting phase separation in the stratum corneum membrane [18,30,31]. Rowat et al. [32] reported that such OA interactions with the stratum corneum were dependent on OA concentration. Generally, OA concentrations as great as 10% are used to achieve an increase of drug skin penetration/permeation [33]. However, in this work, OA was incorporated only at 1% *w*/*w*, as attempts to add greater amounts led to LNPs precipitation 24–48 h after their preparation. Therefore, in addition to OA interactions with the lipid core components, the lack of enhancement effect could be attributed to the low amount of OA used to obtain the investigated LNPs as well. IPM has been reported to exert its enhancement effect by increasing lipid fluidity and/or by promoting drug solubility in the skin [15,34]. In this work, IPM’s ineffectiveness as enhancer could be attributed to its strong interactions with LNPs components of the lipid core that could prevent IPM partitioning and diffusion into the stratum corneum. Other authors outlined a synergic enhancement effect when hydrophilic enhancers such as PG and NMP were used in combination with OA or IPM in conventional topical vehicles [35,36]. Such a synergism was not observed in the present work, likely because OA and IPM were entrapped in the lipid matrix of the LNPs, preventing their free diffusion and interaction with the stratum corneum lamellae. Therefore, further studies have been planned to elucidate the mechanisms involved in OA and IPM interactions with the horny layer when such enhancers are entrapped into LNPs. In particular, studies on IDE-loaded LNPs interactions with a model of biomembrane could be performed, as previously reported [37].

IDE (Figure 5) is a well-known synthetic antioxidant whose beneficial effects after topical application have been widely reported [38,39]. Due to its physicochemical properties such as poor water solubility (3 μg/mL) and high Log *p* value (3.49) [14], IDE effectiveness could be improved by its loading into LNPs. To evaluate the effects on IDE antioxidant activity after its loading into the LNPs under investigation, the ORAC assay, whose results are illustrated in Figure 6, was carried out. After 6 h of incubation, all LNPs showed antioxidant activity greater than that of free IDE, thus highlighting prolonged IDE efficacy due to its incorporation into LNPs.

## 4. Conclusions

Idebenone-loaded LNPs were prepared, including hydrophilic (NMP 10% *w*/*w* or PG 10% *w*/*w*) and/or lipophilic (OA 1% *w*/*w*, IPM 3.5% *w*/*w*, mixture of OA 0.5% *w*/*w* and IPM 2.5% *w*/*w*) penetration enhancers in the water phase or in the lipid core, respectively, to improve idebenone skin targeting. In vitro skin permeation data showed an increased amount of idebenone in the stratum corneum and epidermis when a mixture of OA/IPM was incorporated into LNPs. Unlike what was expected, no synergic effect was observed when lipophilic and hydrophilic penetration enhancers were included simultaneously into LNPs colloidal suspension. Such findings support the hypothesis that skin permeation enhancement mechanisms different from those operating in conventional topical formulations could be involved when hydrophilic and/or lipophilic penetration enhancers are incorporated into LNPs colloidal suspensions. Therefore, the results of this work suggest that the proper choice of chemical penetration enhancers could allow designing topical LNPs formulations with improved drug targeting to the skin.

## Figures and Tables

**Figure 1 pharmaceutics-13-01027-f001:**
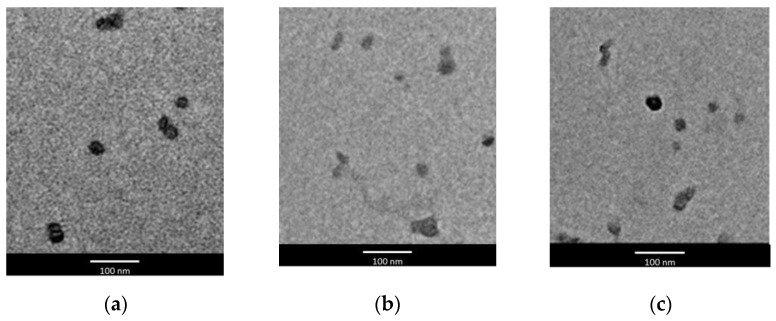
Transmission electron microscopy (TEM) images of idebenone-loaded lipid nanoparticles containing (**a**) isopropyl myristate 3.5% *w*/*w* (formulation BA), (**b**) isopropyl myristate 3.5% *w*/*w* and 10% *w*/*w N*-methylpyrrolidone (formulation BB) and (**c**) isopropyl myristate 3.5% *w*/*w* and 10% *w*/*w* propylene glycol (formulation BC).

**Figure 2 pharmaceutics-13-01027-f002:**
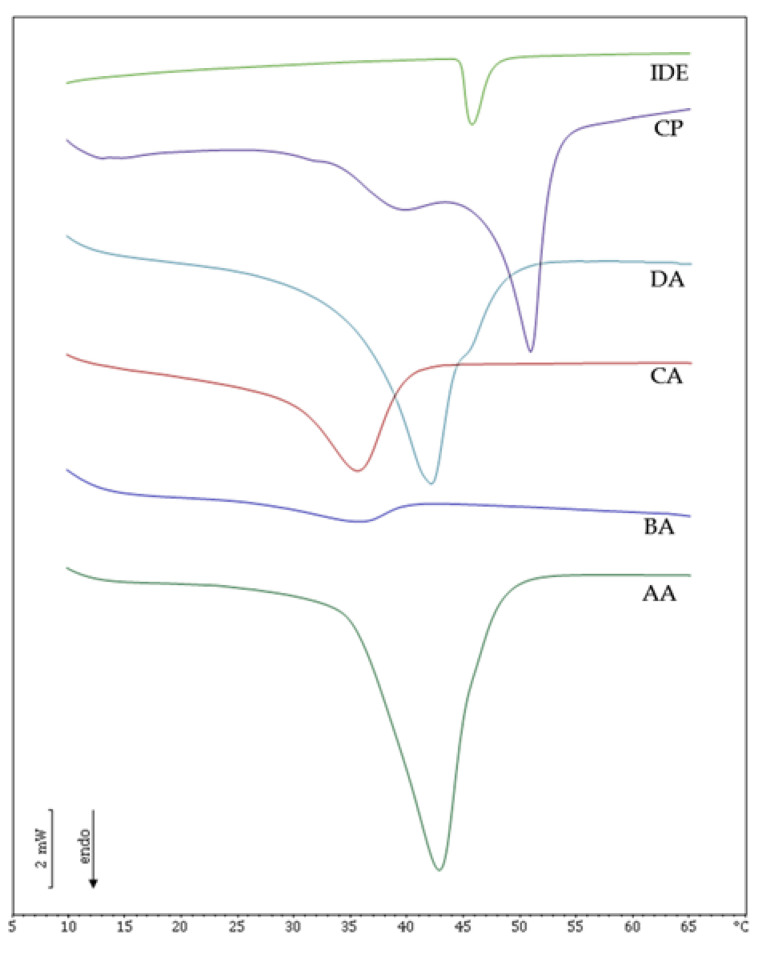
DSC curves of cetyl palmitate (CP), idebenone (IDE) and formulations AA, BA, CA and DA.

**Figure 3 pharmaceutics-13-01027-f003:**
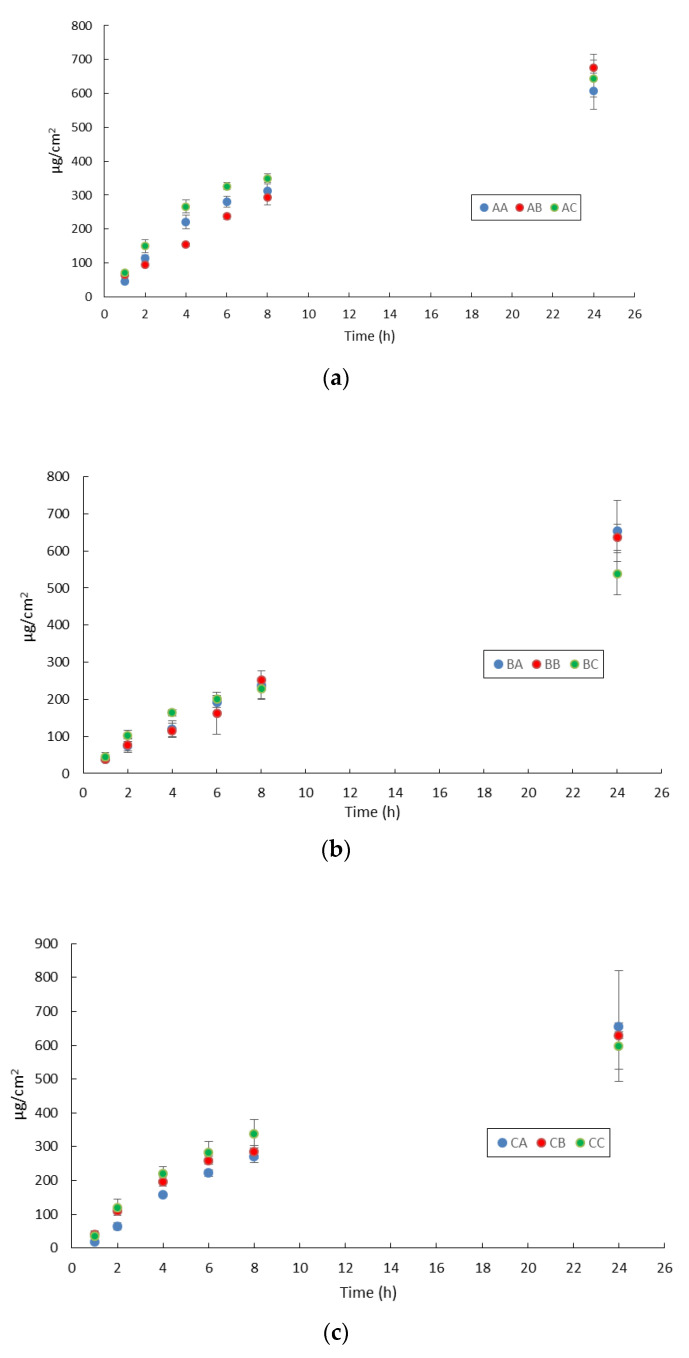
Idebenone release from formulations (**a**) AA, AB, AC; (**b**) BA, BB, BC; (**c**) CA, CB, CC; (**d**) DA, DB, DC.

**Figure 4 pharmaceutics-13-01027-f004:**
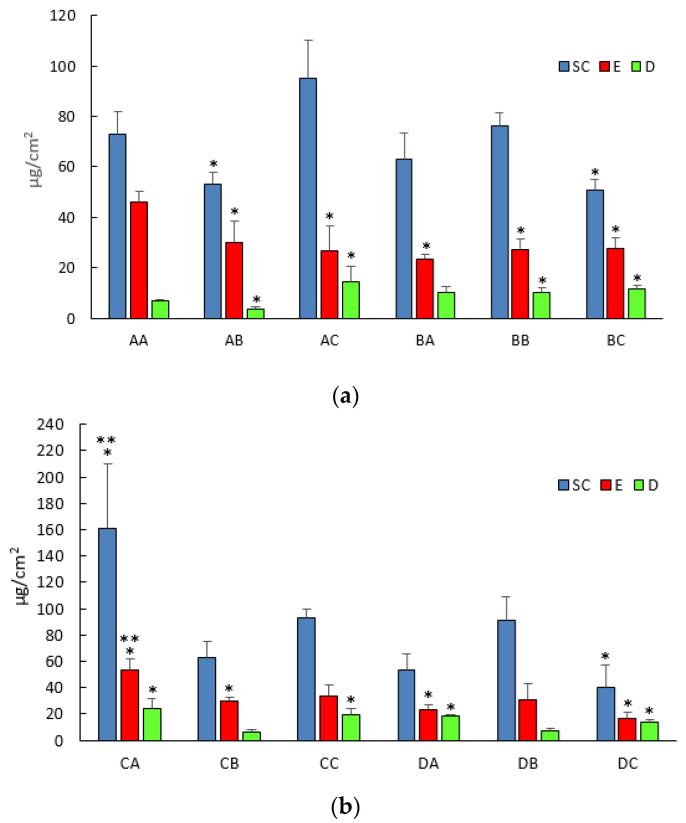
Idebenone skin penetration into stratum corneum (SC), epidermis (E) and dermis (D) from (**a**) formulations AA, AB, AC, BA, BB, BC and (**b**) formulations CA, CB, CC, DA, DB, DC; * statistically different compared to the value of formulation AA for the corresponding skin layer; ** statistically different compared to all other formulations for the corresponding skin layer.

**Figure 5 pharmaceutics-13-01027-f005:**
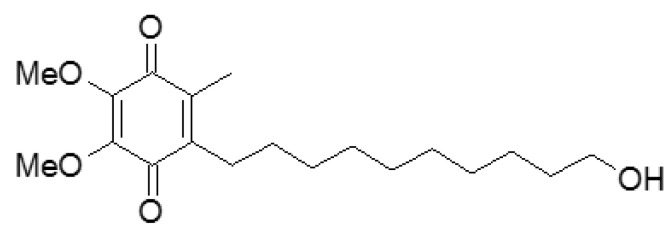
Chemical structure of idebenone.

**Figure 6 pharmaceutics-13-01027-f006:**
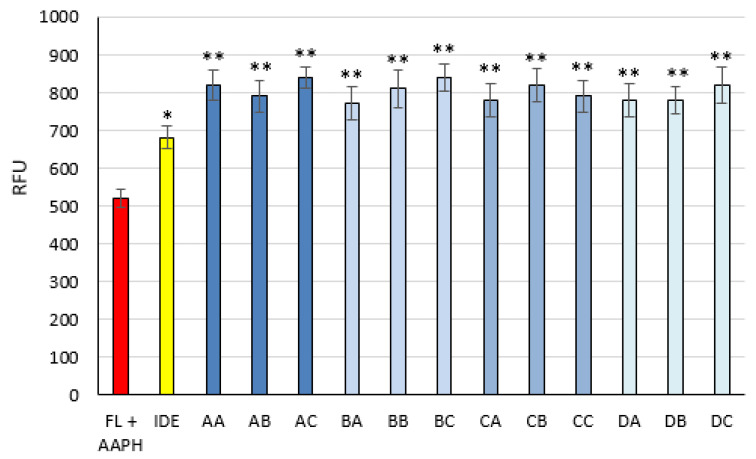
Antioxidant activity of free idebenone (IDE) and formulations AA–DC, expressed as relative fluorescence units (RFU) in the oxygen radical absorbance capacity assay. FL + AAPH = fluorescein + 2,2 azobis (2-aminopropane) dihydrochloride. * Statistically different (*p* < 0.05) compared to FL + AAPH; ** statistically different (*p* < 0.05) compared to free IDE and FL + AAPH.

**Table 1 pharmaceutics-13-01027-t001:** Composition (% *w*/*w*) of formulations AA–DC containing idebenone 0.7% *w*/*w*. GO = glyceryl oleate, CP = cetyl palmitate, IPM = isopropyl myristate, OA = oleic acid, NMP = *N*-methylipyrrolidone, PG = propylene glycol.

Code	Ingredients (% *w*/*w*)							
	Oleth20	GO	CP	IPM	OA	NMP	PG	Water ^1^
AA	8.7	4.4	7.0	---	---	----	---	q.s. 100
AB	8.7	4.4	7.0	---	---	10.0	---	q.s. 100
AC	8.7	4.4	7.0	---	---	---	10.0	q.s. 100
BA	8.7	4.4	3.5	3.5	---	---	---	q.s. 100
BB	8.7	4.4	3.5	3.5	---	10.0	---	q.s. 100
BC	8.7	4.4	3.5	3.5	---	---	10.0	q.s. 100
CA	8.7	4.4	4.0	2.5	0.5	---	---	q.s. 100
CB	8.7	4.4	4.0	2.5	0.5	10.0	---	q.s. 100
CC	8.7	4.4	4.0	2.5	0.5	---	10.0	q.s. 100
DA	8.7	4.4	6.0	---	1.0	---	---	q.s. 100
DB	8.7	4.4	6.0	---	1.0	10.0	---	q.s. 100
DC	8.7	4.4	6.0	---	1.0	---	10.0	q.s. 100

^1^ Water contained Kemipur 100^®^ 0.35% *w*/*w* as preservatives.

**Table 2 pharmaceutics-13-01027-t002:** Lipid nanoparticles’ size (±S.D.), polydispersity index (PDI ± S.D.) and ζ potential (Zeta ± S.D.).

Code	Size ± S.D. (nm)	PDI ± S.D.	Zeta ± S.D. (mV)
AA	29.1 ± 0.2	0.204 ± 0.020	−10.7 ± 1.2
AB	39.5 ± 0.6	0.277 ± 0.006	−9.9 ± 0.3
AC	25.0 ± 0.2	0.182 ± 0.003	−9.7 ± 1.6
BA	19.9 ± 0.3	0.134 ± 0.023	−15.7 ± 1.8
BB	46.8 ± 0.8	0.239 ± 0.003	−7.5 ± 0.8
BC	24.5 ± 0.4	0.191 ± 0.001	−10.3 ± 2.0
CA	32.9 ± 7.5	0.268 ± 0.183	−16.5 ± 2.9
CB	21.2 ± 0.4	0.275 ± 0.028	−12.6 ± 1.1
CC	16.8 ± 0.1	0.135 ± 0.015	−13.2 ± 0.5
DA	60.4 ± 1.9	0.536 ± 0.020	−23.6 ± 0.7
DB	20.7 ± 0.4	0.233 ± 0.001	−16.5 ± 0.3
DC	17.1 ± 0.2	0.141 ± 0.001	−15.7 ± 0.5

**Table 3 pharmaceutics-13-01027-t003:** Mean cumulative amount of idebenone released (Q_released_) or penetrated into the skin (Q_penetrated_) from lipid nanoparticles after 24 h.

Code	Q_released_ (μg/cm^2^)	Q_penetrated_ (μg/cm^2^)
AA	606.73	125.81
AB	675.67	87.02
AC	643.87	136.32
BA	653.77	96.52
BB	635.93	113.44
BC	537.9	90.01
CA	656.24	238.33
CB	630.05	99.3
CC	598.17	142.09
DA	582.87	93.91
DB	637.53	129.12
DC	626.27	70.4

## Data Availability

The data presented in this study are available on request from the corresponding author. The data are not publicly available due to large data sets.

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
