# Peer review of "In Vitro Skin Permeation of Idebenone from Lipid Nanoparticles Containing Chemical Penetration Enhancers"

_pharmaceutics, 2021, doi:10.3390/pharmaceutics13071027_

Round 1

Reviewer 1 Report

Dear Authors,

Your manuscript described the cutaneous penetration of an antioxidant molecule into LNPs prepared with or without enhancers. This issue has merit to be investigated. After careful reading your research material, some questions appeared:

  • corresponding author's contact was not indicated.
  • Abstract lacked the issue context, i.e., a brief introduction.
  • Purity of the active compound was not described in Materials (considering its quantification, it is important to present this data).
  • Table 1 - it was not clear the choice of enhancers' concentrations.
  • Table 1 - please, explain the variations of concentrations from CP, IPM, NMP and PG.
  • Please, provide the active's concentration into the nanostructures for each sample. 
  • considering the alcohol content in the receptor fluid, was the artificial membrane stable in this experiment for 24h?
  • The times for sample collections from the receptor fluid were not registered in Material/Methods.
  • The active in its free form was not tested. Could you justify the absence of this sample?
  • the agitation velocity was not described.
  • were not performed treatments for the active quantification in the strips, epidermis and dermis (lines187-189)?
  • Why was not the diffusion parameters calculated (flux, for instance)? 
  • please, consider to replace standard with analytical in line 202.
  • in lines 230-235, great differences could be noticed among samples. It would be very informative if Authors could speculate such differences that were not correlated with the enhancers.
  • the stability of the samples, including its chemical quantification, is a very important parameter. Please, consider to describe these data.
  • Table 2 lacked statistical treatment. For instance, samples BA and DA were or were not different for size?
  • In TEM imagens, BC formulation appeared 2x. Legend was not self explanatory.
  • in lines 331-341, please, explore more those results and discussion, maybe, applying the diffusion parameters.
  • please, in figure 5, it would be very useful to know the active concentrations in each system to correlate this activity.

Author Response

Dear Authors,

Your manuscript described the cutaneous penetration of an antioxidant molecule into LNPs prepared with or without enhancers. This issue has merit to be investigated. After careful reading your research material, some questions appeared:

  • corresponding author's contact was not indicated.

Answer

We thank the reviewer for the remark. Corresponding author's contact was added at line 11 ([email protected]; Tel.: +39 095 738 4010).

  • Abstract lacked the issue context, i.e., a brief introduction.

Answer

According to the instructions for authors of the Journal Pharmaceutics, the abstract should not exceed 200 words. Therefore, to provide the reader with a clear overview of our work without exceeding 200 words, we could not report more details about the issue context. However, to comply with the reviewer’s request, we added at line 14 the following sentence “As LNPs effectiveness could be increased by the addition of chemical penetration enhancers (PE),” and we modified the abstract not to exceed 200 words as follows:

“Lipid nanoparticles (LNPs) have been proposed as carriers for drug skin delivery and targeting. As LNPs effectiveness could be increased by the addition of chemical penetration enhancers (PE), in this work, the feasibility of incorporating PE into LNPs to improve idebenone (IDE) targeting to the skin was investigated. LNPs loading IDE 0.7% w/w were prepared using hydrophilic (propylene glycol, PG, 10% w/w or N-methylpyrrolidone, NMP, 10% w/w) and/or lipophilic PE (oleic acid, OA, 1% w/w; isopropyl myristate, IPM, 3.5 % w/w; a mixture of 0.5% w/w OA and 2.5 % w/w IPM). All LNPs showed small sizes (<60 nm), low polydispersity index and good stability. According to the obtained results, IDE release from LNPs was not the rate-limiting step in IDE skin penetration. No IDE permeation was observed through excised pigskin from all LNPs while the greatest increase of IDE penetration into the different skin layers was obtained using the mixture OA/IPM. The antioxidant activity of IDE loaded LNPs, determined by the oxygen radical absorbance capacity assay, was greater than that of free IDE. These results suggest that the use of suitable PE as LNPs components could be regarded as a promising strategy to improve drug targeting to the skin.”

  • Purity of the active compound was not described in Materials (considering its quantification, it is important to present this data).

Answer

The purity of idebenone (the active compound) was minum 98% as reported in the compound data sheet. We added this information in the “materials” section at line 70 as follows: 

“Idebenone (IDE, purity: minum 98% as reported in the compound data sheet) was obtained from Carbosynth (Berkshire, UK).”

  • Table 1 - it was not clear the choice of enhancers' concentrations.

Answer

We thank the reviewer for the comment. Regarding the hydrophilic penetration enhancers (NMP and PG), we used them at the same concentration (10% w/w) that is generally regarded as effective for both enhancers.

As reported in the manuscript at line 399, the lipophilic penetration enhancer OA was incorporated at 1% w/w as any attempt to add greater amounts led to LNPs precipitation 24-48 h after their preparation.  The other lipophilic enhancer IPM was used at 3.5% w/w because this was the greatest concentration that provided stable LNPs. Therefore, as far as lipophilic enhancers are concerned, we used the greatest concentration that led to stable LNPs. To provide the reader with such information we added in the text the following sentences

 at line 85:  “NMP and PG were used at a concentration (10% w/w) that is generally regarded as effective for both enhancers [15]”.

at line 87 : “Preliminary experiments were performed to determine the greatest concentration of lipophilic enhancers that provided stable LNPs. According to the results of such experiments, IPM was used at 3.5% w/w while OA was incorporated at 1% w/w. The amount of CP used to prepare LNPs changed to maintain constant the total amount of lipids in the LNPs matrix.”

  • Table 1 - please, explain the variations of concentrations from CP, IPM, NMP and PG.

Answer

As reported in Table 1, NMP and PG were always used at the same concentration (10% w/w). Therefore, we did not report variations of concentrations for NMP and PG.

As mentioned in the sentence we added to explain the choice of penetration enhancers’ concentration, the amount of CP used to prepare LNPs changed to maintain constant the total amount of lipids in the LNPs matrix. This information has been inserted in the text, as mentioned above.

  • Please, provide the active's concentration into the nanostructures for each sample. 

Answer

As reported in the legend of Table 1, each sample contained idebenone 0.7% w/w.

  • considering the alcohol content in the receptor fluid, was the artificial membrane stable in this experiment for 24h?

Answer

We used a receptor fluid containing alcohol to perform in vitro drug release studies through artificial membranes in several previous works (Montenegro L, Turnaturi R, Parenti C, Pasquinucci L. In Vitro Evaluation of Sunscreen Safety: Effects of the Vehicle and Repeated Applications on Skin Permeation from Topical Formulations. Pharmaceutics 10 (2018) pii: E27; Montenegro L., Sinico C., Castangia I., Carbone C., Puglisi G. Idebenone-loaded solid lipid nanoparticles for drug delivery to the skin:  in vitro evaluation. Int. J. Pharm. 434 (2012) 169-174; Montenegro L., Campisi A., Sarpietro M. G., Carbone C., Acquaviva, R., Raciti G., Puglisi G. In vitro evaluation of idebenone-loaded solid lipid nanoparticles for drug delivery to the brain. Drug Dev. Ind. Pharm. 37 (2011) 737-746; Montenegro L., Carbone C., Puglisi G. Vehicle effects on in vitro release and skin permeation of octylmethoxycinnamate from microemulsions. Int. J. Pharm. 405 (2011) 162-168).  In such studies and in the present work, we have never observed any alteration of the artificial membrane due to the use of ethanol in the receptor fluid.

  • The times for sample collections from the receptor fluid were not registered in Material/Methods.

Answer

Samples’ withdrawal of the receiving solution in in vitro release studies was performed at time 0, 1, 2, 4, 6, 8 and 24 h. This information has been added at line 157.

  • The active in its free form was not tested. Could you justify the absence of this sample?

Answer

As reported in literature (Montenegro, L.; Turnaturi, R.; Parenti, C.; Pasquinucci, L. Idebenone: Novel Strategies to Improve Its Systemic and Local Efficacy. Nanomaterials 2018, 8, 87), idebenone (IDE) poor water solubility prevented its formulation in aqueous formulations in its free form. In a previous paper (Montenegro, L.; Sinico, C.; Castangia, I.; Carbone, C. and Puglisi, G. Idebenone-loaded solid lipid nanoparticles for drug delivery to the skin: in vitro evaluation. Int. J. Pharm. 2012, 434, 169-174), we highlighted that vehicles such as oils, creams or nanonemulsions, could have different solubilizing properties, thus affecting IDE thermodynamic activity and making unreliable every comparison between in vitro skin permeation of free IDE from such vehicles and from IDE loaded SLN. Such statement is supported by the work of Li and Ge (Li, B., Ge, Z.Q. Nanostructured lipid carriers improve skin permeation and chemical stability of idebenone. AAPS Pharm. Sci. Technol. 2012, 13, 276–283.), who demonstrated that IDE skin permeation from nanostructured lipid carriers, nanoemulsions or oils depended on vehicle composition.

Therefore, due to the lack of a suitable control vehicle, IDE skin permeation in its free form was not assayed.

  • the agitation velocity was not described.

Answer

The stirring speed (700 rpm) of the receiving solution in in vitro release studies was added at line 153.

  • were not performed treatments for the active quantification in the strips, epidermis and dermis (lines187-189)?

Answer

As reported at line 188, “Tape strips, epidermis, and dermis were placed separately in methanol, sonicated to extract the drug and then the methanol extract was assayed for drug content by HPLC”. Therefore, skin layers were treated with methanol and the resulting samples were sonicated to extract IDE.

  • Why was not the diffusion parameters calculated (flux, for instance)? 

Answer

As reported at line 323-325, “Applying the LNPs under investigation on the skin surface did not provide any IDE permeation as this drug was not detected in the receiving chamber up to 24 h.”

Therefore, as no IDE skin permeation occurred, obviously permeation parameters such as permeability coefficient, diffusion coefficient and flux could not be determined.

  • please, consider to replace standard with analytical in line 202.

Answer

At line 202, the word standard was deleted.

  • in lines 230-235, great differences could be noticed among samples. It would be very informative if Authors could speculate such differences that were not correlated with the enhancers.

Answer

As reported in the text at line 231 (old version), particle sizes ranged from 17 to 60 nm. Therefore, a very small interval of particle size changes was observed. It would a hard task and, likely, a mere academic exercise to formulate a hypothesis supporting such small variations. 

  • the stability of the samples, including its chemical quantification, is a very important parameter. Please, consider to describe these data.

Answer

As reported in the text at line 238 (old version), “all investigated LNPs proved stable during storage for three months at room temperature and sheltered from the light, as no significant change of particles size, PDI and ζ-potential values was detected.” As no particle alteration was observed during the storage period, no chemical modification could be supposed to occur. Therefore, we did not perform any “chemical quantification”.

  • Table 2 lacked statistical treatment. For instance, samples BA and DA were or were not different for size?

Answer

As stated in the introduction, the aim of this work was to investigate the effect of incorporating chemical penetration enhancers into LNPs on in vitro skin permeation/penetration of idebenone from such nanocarriers. Therefore, the investigated LNPs were intended for topical administration and we carefully analyzed all factors that could affect idebenone skin penetration/permeation from such nanocarriers before discussing the results of our work. The data reported in Table 2 clearly highlight that all LNPs containing chemical penetration enhancers showed sizes that were greater or smaller than the LNPs formulation (formulation AA) without chemical penetration enhancers. In addition, apart from formulation AC and BC, all LNPs showed sizes that were statistically different from each other (Student’s t-test: p<0.05 for all comparisons apart from AC vs BC). As can be noted analyzing the data in Table 2, a very small interval of particle size changes was observed as particle sizes ranged from 17 to 60 nm. Such small particle size changes, although statistically significant, cannot be expected to affect the skin penetration/permeation process or the release of the drug loaded in these LNPs as all nanoparticles had small sizes. The effect of LNPs size on drug skin permeation/penetration could be taken into account for nanoparticles showing bigger sizes (i.e. > 100-200 nm). Consequently, the different IDE skin penetration from the LNPs under investigation cannot be attributed to their different sizes. Therefore, we believe that a statistical analysis of the differences among particle size is not relevant in the discussion of our results being the changes of particle sizes too small to influence the drug permeation/penetration process.

  • In TEM imagens, BC formulation appeared 2x. Legend was not self explanatory.

Answer

We apologize for this typo. To be self-explanatory, the legend of Fig. 1 has been amended as follows: “Transmission electron microscopy (TEM) images of idebenone loaded lipid nanoparticles containing (a) isopropyl myristate 3.5% w/w (formulation BA), (b) isopropyl myristate 3.5% w/w and 10% w/w N-methylpyrrolidone (formulation BB), and (c) isopropyl myristate 3.5% w/w  and 10% w/w propylene glycol (formulation BC).

  • in lines 331-341, please, explore more those results and discussion, maybe, applying the diffusion parameters.

Answer

As mentioned above, no IDE skin permeation occurred. Therefore, “diffusion parameters” could not be determined and could not be used to discuss the reported results. The obtained data do not support any further speculation that could be regarded as convincing and reliable.

  • please, in figure 5, it would be very useful to know the active concentrations in each system to correlate this activity.

Answer

All LNPs contained the same amount of active ingredient (IDE 0.7 % w/w) and the same dilution for all LNPs sample was used. Free IDE concentration in the sample was the same contained in the diluted IDE loaded LNPs samples. 

Reviewer 2 Report

Ms.Nr: pharmaceutics-1269880

Title: In vitro skin permeation of idebenone from lipid nanoparticles containing chemical penetration enhancers

Manuscripts describes a comprehensive experimental work studying the skin penetration/permeation of idebenone antioxidant drug loaded into LPNs. The effect of chemical penetration enhancers (both hydrophilic and lipophilic) on in vitro release and on in vitro permeation through pig skin of the model compound from 12 LNP formulations was investigated using Franz cell method. The topic is current regarding the growing interest of nanoparticles as delivery systems for topical administration and the use of different penetration enhancers to improve drug skin targeting and delivery. The experimental work is well design and conducted. The construction of the manuscript is logic, clear and well-readable. However, there are some shortcomings of the paper what should be addressed before accepting for publication. I suggest the acceptance after revision.

Critical remarks:

  1. In the experiments infinite doses (200microL) of LNP preparations were used and 24h incubation time was applied. I do not think to be relevant the 24h residence on the skin in real human application. As all in vitro method is aimed to predict the in vivo performance, authors should explain why this long incubation is used.
  2. The basic chemical information about idebenone has to be inserted (structure, physico-chemical properties, etc.) Particularly important question is the thermal-stability of the compound since the preparation of LPNs by PIT method heats the phases up to 90 degree of C. The stability of IDE at this temperature has been controlled?
  3. On Fig. 3 the IDE release profiles are shown. From 0-8h period of time, the points fit rather to exponential than linear curve. Because of no measured data between 8 and 24h, to use linear regression is nonsense. Authors should not draw from these data as a conclusion that the release being pseudo-first order kinetic. Reviewer suggestion is to remove the linear curve and to enlarge the figure in 0-8h period. Enough that the cumulative results (Qreleased) are indicated in Table 3.
  4. The interpretation of Qpenetrated results (Table 3) in the text (lines 330-345) contains contradictions. Authors have to check the conclusions in this section about BA and CC formulations.
  5. There is a misprint in legend of Fig. 1. images (b) formulation is BB; and in line 72, in the chemical name of NMP.

Author Response

As the copy and paste option does not allow us inserting plots, we uploaded our answers's to the reviewer's comments as PDF file.

Reviewer 3 Report

The paper is well written and provide an interesting contribution to the wide area of lipid nanoparticles as drug carriers, specifically at how small amounts of modifiers can change the skin penetration of a loaded drug.

In Page 2 rows 50-51 the authors state “to date few data has been reported about the ability of chemical penetration enhancers to improve drug skin penetration/permeation from LNPs”:

the authors should cite them (or part of them)

PAGE 6 "all investigated LNPs proved stable during storage for three months at room temperature and sheltered from the light, as no significant change of particles size, PDI and ζ-potential values was detected (data not  shown)"
PAGE 7 "The addition of hydrophilic enhancers (PG or  NMP) to the aqueous phase of formulations AA, BA, CA, and DA did not led to any significant change of the thermal behavior of the resulting colloidal suspensions (data not shown)":
  the data should be included as supplementary material

Author Response

The paper is well written and provide an interesting contribution to the wide area of lipid nanoparticles as drug carriers, specifically at how small amounts of modifiers can change the skin penetration of a loaded drug.

In Page 2 rows 50-51 the authors state “to date few data has been reported about the ability of chemical penetration enhancers to improve drug skin penetration/permeation from LNPs”:

the authors should cite them (or part of them)

Answer

We thank the reviewer for the comment. We inserted in the text, at line 52, the following sentences citing some works on the use of chemical penetration enhancers in association to LNPs.

Transcutol P, a hydrophilic chemical penetration enhancer, has been reported to increase skin accumulation of 8-methoxypsoralen from LNPs [12].  Patel et al. [13] added lipophilic penetration enhancers (d-limonene and oleic acid) to gel formulations containing raloxifene loaded solid lipid nanoparticles showing that the addition of d-limonene provided an increase of drug permeation through excised human skin.

The following references were added to the references’ list:

  1. Pitzanti, G.; Rosa, A.; Nieddu, M.; Valenti, D.; Pireddu, R.; Lai, F.; Cardia, M.C.; Fadda, A.M.; Sinico, C. Transcutol® P Containing SLNs for Improving 8-Methoxypsoralen Skin Delivery. Pharmaceutics 2020, 12, 973. https://doi.org/10.3390/pharmaceutics12100973.
  2. Patel, K.K.; Gade, S.; Anjum, M.M.; Singh, S.K.; Maiti, P.; Agrawal, A.K.; Singh, S. Effect of penetration enhancers and amorphization on transdermal permeation flux of raloxifene-encapsulated solid lipid nanoparticles: an ex vivo study on human skin. Nanosci. 2019, 9, 1383–1394. https://doi.org/10.1007/s13204-019-01004-6.

PAGE 6 "all investigated LNPs proved stable during storage for three months at room temperature and sheltered from the light, as no significant change of particles size, PDI and ζ-potential values was detected (data not  shown)"
PAGE 7 "The addition of hydrophilic enhancers (PG or  NMP) to the aqueous phase of formulations AA, BA, CA, and DA did not led to any significant change of the thermal behavior of the resulting colloidal suspensions (data not shown)":
  the data should be included as supplementary material

Answer

We decided not to report such data to avoid inserting tables or plots illustrating results that did not provide the reader with useful information. Thermograms of LNPs containing PG or NMP were almost superimposable to those of LNPs without these enhancers, while tables or plots reporting stability data would be a mere repetition of many values so similar that the reader would not like wasting time to read them.

Therefore, we believe that it is not appropriate to report these data even as supplementary material in prestigious journals such as Pharmaceutics.

Round 2

Reviewer 1 Report

Dear Authors,

Thank you for addressing all questions.

Reviewer 2 Report

As the authors made the necessary corrections and gave sufficient response to the questions, I suggest the revised manuscript for publication.